# The Role of Early Life Microbiota Composition in the Development of Allergic Diseases

**DOI:** 10.3390/microorganisms10061190

**Published:** 2022-06-09

**Authors:** Maimaiti Tuniyazi, Shuang Li, Xiaoyu Hu, Yunhe Fu, Naisheng Zhang

**Affiliations:** Department of Clinical Veterinary Medicine, College of Veterinary Medicine, Jilin University, Changchun 130062, China; mmttn18@mails.jlu.edu.cn (M.T.); lishuang@mails.edu.jlu.cn (S.L.); huxiaoyu@mails.jlu.edu.cn (X.H.)

**Keywords:** allergic diseases, microbiota, early life

## Abstract

Allergic diseases are becoming a major healthcare issue in many developed nations, where living environment and lifestyle are most predominantly distinct. Such differences include urbanized, industrialized living environments, overused hygiene products, antibiotics, stationary lifestyle, and fast-food-based diets, which tend to reduce microbial diversity and lead to impaired immune protection, which further increase the development of allergic diseases. At the same time, studies have also shown that modulating a microbiocidal community can ameliorate allergic symptoms. Therefore, in this paper, we aimed to review recent findings on the potential role of human microbiota in the gastrointestinal tract, surface of skin, and respiratory tract in the development of allergic diseases. Furthermore, we addressed a potential therapeutic or even preventive strategy for such allergic diseases by modulating human microbial composition.

## 1. Introduction

In recent decades, allergic diseases such as asthma, atopic dermatitis (AD), and food allergy (FA) have become a major healthcare issue in many Western countries and developed nations around the world [1,2]. Studies have identified that genetic and environmental factors are the main causes of allergic diseases [3,4]; the topic, however, needs more exploration.

Allergy is positively related to the degree of development of human society, where living environment and lifestyle are most predominantly distinct [5,6]. Such changes include urbanized and industrialized living environments with overused hygiene products and antibiotics, coupled with stationary lifestyles and fast-food-based diets. All such factors result in reduced microbial diversity in early life [7], which, according to ecosystem theory, leads to impaired immune protection and to slow recovery of normal microbial communities [8].

Human microbiota has become an increasingly popular area of study due to its role in host physical and mental health and metabolism [9,10]; it comprises of bacteria, viruses, fungi, protozoa, and archaea. Although human microbiota can be found in the oral and nasal cavity, on surfaces of the skin, and in the respiratory and reproductive tracts [11], it primarily colonizes the gastrointestinal tract. Accumulating evidence indicates that human microbiota plays an important role in the development and prevention of allergic diseases [12]. Commensal microbial communities in the gastrointestinal tract and other organs have shown to modulate both innate and acquired immune responses via various axes, including the gut–lung axis and the gut–skin axis. Recent studies showed that numerous environmental factors can affect microbiota colonization, composition, and metabolic activity in early life, and modify the host functions digestion and nutrient absorption for host energy production and immune modulation and protection [13,14,15,16]. Naturally, early life microbiota colonization with healthy microbiota with proper diversity and abundance is a critical factor in the later development of immune protection in infants. In contrast, dysbiosis of the microbial community is associated with greater disease susceptibility and immune-related disorders later in life, including allergic diseases [12,17,18].

Bacteria, previously, were considered as pathogens; however, it is evident that they have a crucial role in host physiology [19]. Recent advances in culture-independent DNA-sequencing technology (i.e., 16S rRNA sequencing) and data analysis methods have revealed that every part of the human body is colonized with different microbial species, which play a complex role in the pathogenesis of FA [20], AD [21], and asthma [22]. In this paper, we aimed to review recent findings on the potential role of the human microbial community in the gastrointestinal tract, the surface of the skin, and the respiratory tract in the development of allergic diseases. Furthermore, we addressed a potential therapeutic or even preventive strategy for such allergic diseases by modulating human microbial composition.

## 2. Maternal Influencing Factors for the Development of Allergic Diseases in Infants

Numerous studies have shown that microorganism colonization begins in mammals during birth, and its composition can be influenced by several prenatal and postnatal environmental and host-related factors that have vital roles in the development of a healthy immune system. Among such factors (Figure 1), delivery methods (vaginal or cesarean section delivery) [23,24,25], feeding choices (breast or bottle feed) [26], antibiotic or probiotic use [27,28], and other processes of early gut microbiota modulation by vaginal fluid or fecal microbiota transplantation (FMT) [29,30,31] can dramatically change the gut microbiota composition and modulate the infant’s immune development and tolerance to different antigens. Delivery mode determines the colonization of early life microbiota in infants. For example, babies born by cesarean section lack commensal microbial communities that can be found in vaginal-born infants [32]. Instead, such delivery approaches result in colonization of pathogenic bacteria such as *Enterococcus*, *Enterobacter*, and *Klebsiella* species that are typically found in the hospital environment [33]. Although such a microbial gap is mainly closed after 6 to 9 months of breastfeeding (except for *Bacteroides*, which remain absent or at a very low level in most cesarean-section infants), cesarean-section delivery can increase the susceptibility of respiratory infectious disease in the first year of life, which is determined by the first week of microbial colonization [34].

Only naturally born infants have gut microbiota similar to their mother’s vaginal microbial community; infants born by cesarean section have gut microbiota similar to their mother’s skin microbiota instead [48]. For example, at an early age, infants have microbiota that are similar to their mother’s vaginal microbial community which mainly consist of *Lactobacillus species*, suggesting that the infants might have obtained a certain part of their microbiota from the birth canal during birth [49]. However, it has also been reported that *Lactobacillus* and *Streptococci* are found in high numbers in a mother’s milk [50], indicating breastfeeding has a significant impact on an infant’s gut microbial composition. Weaning (breastmilk) plays a role as an additional inoculum of the infant gut [51], which not only has microbes but also enriches bacterial species such as *Bifidobacterium* (utilizing nutrients in breastmilk). These bacteria, as the first arrivals in the infant intestinal tract, consume all the oxygen and create a suitable anaerobic condition for further colonization by other species that are characteristic in the healthy adult gut microbial community. Microbial communities in babies that are born vaginally and fed with breast milk are termed as having relatively healthier microbiomes with the highest abundance of *Bifidobacteria* and the lowest number of opportunistic pathogenic bacteria such as *Clostridium difficile* and *Escherichia coli* [50,52,53,54,55].

Early life antibiotic use both in pregnancy and the postnatal stage influences the establishment of normal infant gut microbiota and increases the development of allergic diseases [56,57]. Infants from mothers exposed to antibiotics during delivery showed decreased microbial diversity compared to non-exposed infants. The microbiota of infants exposed to antibiotics was characterized by a decreased abundance of *Bacteroidetes* and *Bifidobacteria*, with a concurrent increase of *Proteobacteria*, which were most pronounced in terms of vaginally born infants. Furthermore, antibiotics administered during pregnancy and labor have been associated with an elevated risk of AD [58] and asthma [59,60].

Probiotics, vaginal, and/or fecal microbiota transplantation are three major methods to modulate early life gut microbiota in infants and result in favorable outcomes, especially in the preventive effect on disease development that may occur later in life [30,31,61]. However, it is worth mentioning that such microbial modulation therapies are a time-sensitive issue. According to previous reports [62,63], the period of the first 1000 days of an infant’s life, beginning from conception to 2 years of age, is a vital window of opportunity for microbiota modulation. The infant gut microbiota becomes more mature and individual both in functions and compositions after this period. Later in life, gut microbiota are mostly influenced by factors including antibiotic or probiotic, dietary change, and FMT, and all of them can alter immune responses, thereby changing the host’s ability to defend against diseases including allergic, infectious, and autoimmune disorders.

## 3. Non-Maternal Influencing Factors for the Development of Allergic Diseases in Infants

The infant’s gut microbiota is relatively much less populated compared to adults [64], and its initial composition greatly affects the host in terms of whether the host could develop proper immune responses to protect from various diseases later in life (Figure 2). A study involving 14,572 children [65], among whom 10,220 received at least one antibiotic treatment during the first 2 years of life, showed that early antibiotic exposure was associated with an increase risk of childhood asthma, allergic rhinitis, atopic dermatitis, celiac disease, overweight, obesity, and attention deficit hyperactivity disorder. Although such links are also influenced by the quantity, type, and timing of antibiotic exposure, any disruptions of gut microbiota result in increased susceptibility to various disorders. Germ-free experimental animals are best for studying the role of gut microbiota. Such studies have proved that there is a codependent relationship between gut microbiota and immune system development [66,67,68,69,70,71,72,73,74,75,76].

In human studies, in terms of allergic diseases, gut microbiota showed a vital role in the establishment of adaptive and innate immunity protection. For example, compared to healthy infants, babies with lower IgG responses to specific clusters of microbiota antigens are closely related to the development of allergic diseases including asthma, AD, and FD [12,85,86]. Studies have shown that infants with high risk of AD are associated with lower abundance of *Proteobacteria* with increased toll-like receptor (TLR)-4-induced innate inflammatory responses, while depletion of *Ruminococcaceae* is associated with increased TLR-2-induced innate inflammatory responses [87,88,89]. In recent years, the role of gut microbiota in asthma has become a popular area of study. Indeed, infants that are at a greater rate of developing asthma have lower abundance of some gut bacterial taxa such as *Faecalibacterium* and *Bifidobacterium* [90,91]. Similarly, food allergy in early age is also closely related to reduced gut microbial abundance [12]. Such studies suggest that modulating gut microbiota to a normal composition with proper abundance and function may be a novel method for promoting regulatory tolerogenic immune responses.

## 4. The Role of Lung and Gut Microbiota in Asthma

Asthma is one of the most serious allergic diseases both in children and adults in the developed world, currently affecting 300 million people—a number that is increasing every year [22]. Its connection to the gut microbiota was established years ago, and studies have indicated that early-life antibiotic exposure, diet, formula feeding, cesarean section, and an industrialized living environment that are directly involved in altering gut microbiota could aggravate asthma. Especially in the first year of life, when the maturation of gut microbiota occurs, any disruptions during this period of development may cause asthma and other immunological diseases [92,93,94,95,96,97,98,99,100,101,102,103,104,105]. According to a previous study [90], gut microbiota of neonates is closely related to development of allergic diseases; the lowest relative abundance of *Bifidobacteria*, *Akkermansia*, and *Faecalibacterium* genera and higher relative abundance of *Candia* and *Rhodotorula* fungi have the highest risk of developing atopy and asthma. Therefore, such data suggest that the complex and dynamic nature of the gut microbiota may be an important factor in the development of asthma symptoms.

In addition to gut microbiota, mounting evidence suggests that the lung microbiota is also involved in the onset of respiratory diseases, especially in early life [106,107,108,109]. This connection is supported by not only preclinical trials but also case-controlled animal experiments [110,111,112].

Healthy lungs are predominantly colonized with commensal bacterial phylum such as *Bacteroides* and *Prevotella* spp. [113,114]. Similar to gut microbiota, lung microbiota has a critical period of 2 weeks, during which it promotes the transient expression of programmed death ligand 1 (PDL1) in dendritic cells, which is vital for the Treg-mediated attenuation of allergic airway responses [115]. Exposing children to a diverse microbial environment is important for establishing a healthy immune response. Studies have shown that children who grow up in farms [82,103,116], where they have much more contact with microorganisms compared to urban environments, have a lower rate of developing allergic diseases. Studies have also indicated that early life respiratory tract colonization with certain bacteria, such as *Streptococcus*, *Moraxella*, or *Haemophilus*, increase the severity of lower respiratory viral infection in the first year of life, and the risk of developing asthma symptoms later in life [117].

Asthma is not a single disease. It is a result of a complex interaction which involves two major elements: the mother and the baby (Figure 3). Each of them has an individual or combined contribution to the development of asthma. On the other hand, such complexity also creates more opportunity for treating and preventing asthma during pregnancy and early life by various approaches. For example, antibiotic use during pregnancy increases asthma susceptibility in children; however, the severity of asthma may depend on the dose, type, and timing of their usage [35,39,65,118,119,120]. Such human studies are also proven in animal experiments [121]. Therefore, careful use of antibiotics during pregnancy could alleviate asthma symptoms in the offspring. After delivery, during the window of opportunity, modulating gut microbiota via different methods, including probiotic supplement [121,122,123,124], fecal, or vaginal microbiota transplantation [31,125,126], can ameliorate asthma in children. From the standpoint of exposing infants to diverse microbial communities, raising children in a farming environment, a big family, or with pets could also increase tolerance of allergens, thereby decreasing allergic diseases including asthma.

At the same time, the most direct approach may be supplementing short chain fatty acid (SCFA), which can promote the maturation of dendritic cells in bone marrow, leading to mature cells with reduced ability to instigate Th2 responses in the lung and to induce IgA production by mucosal B cells [130]. SCFAs, especially butyrate acid produced by dietary fiber in the presence of *Faecalibacterium prausnitzii* [131], have an anti-inflammatory role and can promote epithelial barrier permeability.

## 5. The Role of Skin and Gut Microbiota in Atopic Dermatitis

AD, a chronic inflammatory skin disease, is also a major issue we are facing in modern times; it affects 15–30% of children and 10% of adults [132]. Its pathogenesis remains obscure [133], but it is considered to be a result of a complex combination of the immune response, the impaired barrier function, and microbiota elements. Among those factors, skin and gut microbiota seem to be more directly related to the development of AD (Figure 4). Studies showed that changes in skin microbiota immune modulation are due to disturbances in epidermal barrier function [134]. Skin microbiota composition is mainly influenced by age, gender, ethnicity, climate, ultraviolet exposure, and lifestyle [135]. Healthy skin surface is colonized with commensal bacterial species [136], such as *Lipophilic Propionibacterium* species, *Staphylococcus* and *Corynebacterium* species.

AD is a complex skin disorder resulting from epidermal barrier dysfunction, altered innate/adaptive immune responses and impaired skin microbial biodiversity [143]. Indeed, healthy skin microbiota protects the surface from various diseases including acute and chronic AD. When the skin microbiota loses microbial diversity [137], with the predominance of the *Staphylococcus* aureus over *Staphylococcus epidermidis*, AD occurs. Studies also showed that skin microbiota diversity is also related to AD and the risk of allergic sensitization to common allergens [144].

Similar to gut and lung microbiota, the composition of skin microbiota at an early age is also related to AD. For instance, a study showed that 2-month-old babies with lower abundance of *Staphylococci species* on their skin had a lower risk of developing AD at 1 year [145]. This is due to early life colonization of the skin by *Staphylococci epidermidis* being associated with the induction of specific Tregs that modulate activation of host immune responses locally [146].

Interestingly, unlike other allergic diseases, AD is not or is poorly associated with cesarean delivery [147,148,149]. Such data further indicate the role of skin microbiota in the development of AD. Studies showed that, in the hospital environment, especially in the operating room, bacteria such as *Staphylococcus* and *Corynebacterium* predominate [150], which are healthy skin microbiota. Therefore, first connecting with healthy skin microbiota could act as a shield for resisting colonization by bacteria that may induce AD. Based on such results, skin microbiota modulation via probiotics or healthy skin microbiota may provide us a novel therapeutic approach for alleviating AD symptoms [151].

Recent studies indicated that gut microbiota is associated with immune modulation as a factor of AD development [152,153]. Data showed that the severity of AD is closely related to the abundance of certain bacteria. For example, a study indicated that, compared to healthy controls, people with AD have a lower density of *Bifidobacterium* in their intestinal tract [139]. However, the count and percentage of *Bifidobacterium* is different according to the stage of AD. Early gut microbiota colonization is associated with various diseases, including AD. For example, *Clostridium difficile* is related to the development of AD, while lower abundance of *Bacteroidetes* at 1 month of age is associated with AD at 2 years of age [154,155,156]. A recent study showed that, compared to healthy school children, the gut of patients with AD was significantly less abundant in some bacterial species, namely *Lachnobacterium* and *Faecalibacterium* [127]. Such studies highlight the possibility of preventing and treating AD by modulating gut microbiota. Indeed, evidence has suggested that oral supplementation of *Lactobacillus* and *Bifidobacterium* strains could reduce the risk of AD in infants by regulating T cell-mediated responses [157]. FMT, as the most direct approach of modulating gut microbiota, is reported to be associated with suppression of AD-induced allergic responses by restoration of gut microbiota and immunological balance both in human and animal studies [158,159].

## 6. The Role of (Oral and) Gut Microbiota in Food Allergy

It is obvious that oral and gut microbiomes are closely related to food allergies (Figure 5). The oral mucosa is the first entity to come into contact with antigens and is the beginning of a continuous gastrointestinal ecosystem that contains local antigen-presenting cells and lymphoid cells, and is associated with organizing lymphoid structures [160]. As studies evidence by subclinical immunotherapies [161], antigen exposure and presentation by oral immune cells modulate systemic immune tolerance. The oral cavity is colonized by a complex microbial community that is directly connected to the gut microbiota both in early life and during pathogenic reaction [162,163]. The composition of the oral microbiota is influenced by birth mode and parents. Data showed that the composition of oral microbiota has distinct colonization patterns between cesarean section and vaginally delivered infants with vaginally born babies having a higher number of taxa [164]. In addition, a recent study found that during the first 18 months of life the oral microbiota of infants was influenced by their parents and shared commensal and disease-related bacteria [165]. This may be why breastfeeding and exposure to diverse microbial environments such as farms and big homes are important for decreasing the incidence of allergic diseases in infants [166,167]. At the same time, a study showed that residential microbial communities favor the crosstalk between innate myeloid and lymphoid cells that contributes to immune homeostasis in the gut and the development of oral tolerances to oral antigens [168].

From the perspective of gut microbiota, a previous study indicated that infants with cow’s milk allergy (CMA) had relatively higher abundant bacterial taxa, particularly anaerobics, compared to healthy controls after 6 months of milk formula feeding [172]. More precisely, according to this study, gut microbiota of infants with CMA had higher concentrations of *Lactobacilli* and lower concentrations of *Enterobacteria* and *Bifidobacteria*. Infants whose CMA was resolved by 8 years of age had an enhanced *Clostridia* and *Firmicutes* rate in their gut [173]. The gut microbiota of children with egg allergy had a greater abundance of certain genera compared to healthy ones, namely *Lachnospiraceae* and *Ruminococcaceae* [174]. A recent study involving 14 children with food allergy and 87 children with food allergen sensitization found that *Dorea*, *Haemophilus*, *Dialister*, and *Clostridium genera* were reduced in healthy participants, while the *genera Citrobacter*, *Lactococcus*, *Oscillospira*, and *Dorea* were reduced in participants with food allergy [175]. Furthermore, data have shown that, compared to healthy controls, the gut microbiota of peanut or tree nut allergy patients had a decreased richness and increased concentration of *Bacteroides* species [176].

Germ-free mice are ideal for studying gut microbiota-related human diseases. Indeed, the role of gut microbiota in the development of CMA has been proven to be prominent. A study showed that germ-free mice were protected from developing susceptibility to CMA if colonized with gut microbiota from healthy infants [177]. Furthermore, transferring specific bacterial strains, *Bifidobacterium* or *Clostridium*, to mice was shown to reduce the risk of food sensitization by inducing mucosal Treg [178]. A study also showed that *Clostridia* can stimulate innate lymphoid cells to produce IL-22, which contributes to straightening the epithelial barrier and decreasing the permeability of the intestine to dietary proteins [179]. Some functional effects of *Clostridia* in food allergy may also exert, via their fermentation metabolites such as butyrate, an SCFA that is known for its role in immunoregulatory and tolerogenic properties [180,181,182,183,184,185]. In addition, butyrate is the only SCFA produced exclusively by gut microbial fermentation, while others are influenced by host metabolism [186]. Recent findings support the hypothesis that butyrate might contribute to the development of immune oral tolerance and the prevention and treatment of food allergies [187,188,189].

Previous human and animal studies established the association between oral and gut microbiota and food allergy. Although the mechanisms involved are complex and dynamic, they underline the possibility of preventing and treating food allergy by microbial modulation. For example, a study showed that, compared to non-supplemented hypoallergenic milk formula, supplementation with hydrolyzed casein formula containing the probiotic *Lactobacillus rhamnosus GG* promoted CMA resolution at 12, 24, and 36 months [190], which was found to enrich butyrate-producing bacterial strains [180]. Using an amino-acid-based formula that contained a specific synbiotics, a combination of prebiotic blend of fructooligosaccharides and the probiotic strain *Bifidobacterium breve M-16V* has been shown to modulate the gut microbiota and its metabolic activities in infants with non-IgE-mediated CMA [191,192,193]. In addition, a study indicated that oral supplementation with *Lactobacillus rhamnosus GG* could enhance the efficacy of oral immunotherapy in inducing peanut tolerance and immune changes in children with peanut allergy [194]. However, this was an uncontrolled study; future studies including a control group are needed to further determine such results.

Fecal microbiota transplantation is another potential therapeutic approach for food allergy. Studies showed that re-establishing the gut microbiota of patients can ameliorate the allergic symptoms by increasing microbiota diversity in FMT trials [195,196]. Dysbiosis of gut microbiota leads to development of food allergy [173]; then, restoration of immune homeostasis and reconstruction of the impaired gut microbiota barrier by FMT may be able to promote the development of oral tolerance [195].

## 7. Conclusions

Microbiota during pregnancy and an infant’s early life are both crucial for the development of a healthy immune system and disease protection. Inappropriate or insufficient microbiota, either in mothers or infants, can have various harmful effects on immune health, contributing to the development of allergic diseases. Although recent studies have deepened our understanding of the relationships between maternal and infant microbiota and the immune system with respect to allergic diseases, the mechanisms involved at molecular levels remain unelucidated. Indeed, allergic disease is not a single disease but rather a result of complex microbial and immune interactions involving both the mother and the infant. Such complexity gives us opportunities for intervention and modulation both during pregnancy and early infant life to decrease allergic symptoms. Therefore, understanding the mechanisms involved is of utmost importance for developing effective and safe prevention strategies for allergic diseases.

## Figures and Tables

**Figure 1 microorganisms-10-01190-f001:**
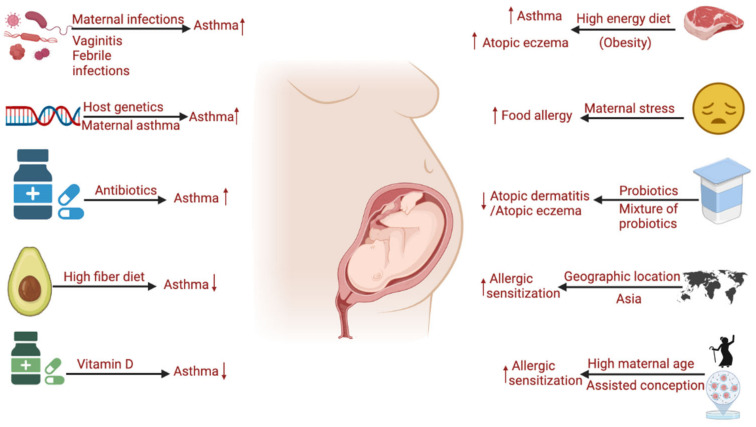
Maternal influencing factors for development of allergic diseases in infants. Maternal infectious diseases [35,36], asthma [37], antibiotic exposure [38,39], and high fat (energy) diet increase the risk of asthma in infants. High fiber diet [40], and Vitamin D [41] supplement during pregnancy could decrease the rate of asthma in children. Maternal stress [42,43] and high age [44] contribute to the development of food allergy in infants. High-energy diet during pregnancy increases the risk of AD in infants [45], while probiotics or a mixture of probiotics protects infants from AD risk [46]. High maternal age and certain geographic location (i.e., Asia) are closely related to increased infant allergic sensitization [47]. (Arrows, upward: increased risk for allergic diseases; downward: decreased risk for allergic diseases). (Created with BioRender.com).

**Figure 2 microorganisms-10-01190-f002:**
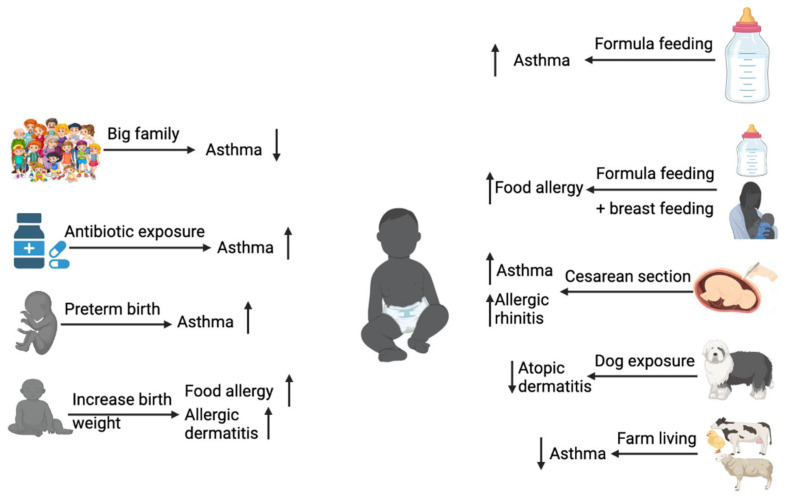
Influencing factors in the development of allergic diseases in infancy. Factors such as antibiotic exposure [77], preterm birth [78], formula feeding [79], and cesarean section delivery [80] increase the development of asthma in infants, while living in a big family [81] and farming environment [82] decrease asthma incidents. Increased birth weight [83] and combined feeding of breast milk and formula [84] increase food allergy development in childhood. (Arrows, upward: increased risk for allergic diseases; downward: decreased risk for allergic diseases). (Created with BioRender.com).

**Figure 3 microorganisms-10-01190-f003:**
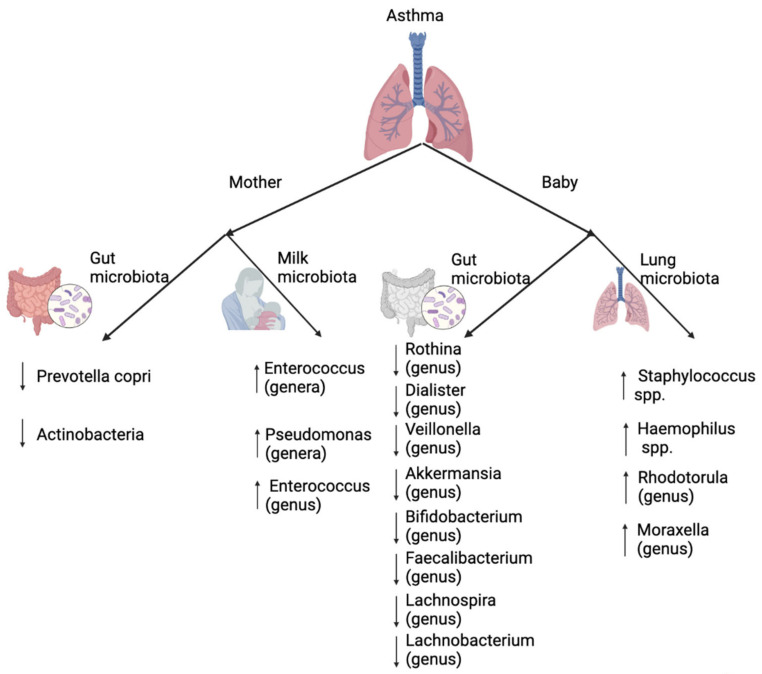
The role of maternal gut and milk microbiota and infant gut and lung microbiota in the development of asthma [59,91,105,117,127,128,129]. (Arrows, upward: increased relative abundance; downward: decreased relative abundance). (Created with BioRender.com).

**Figure 4 microorganisms-10-01190-f004:**
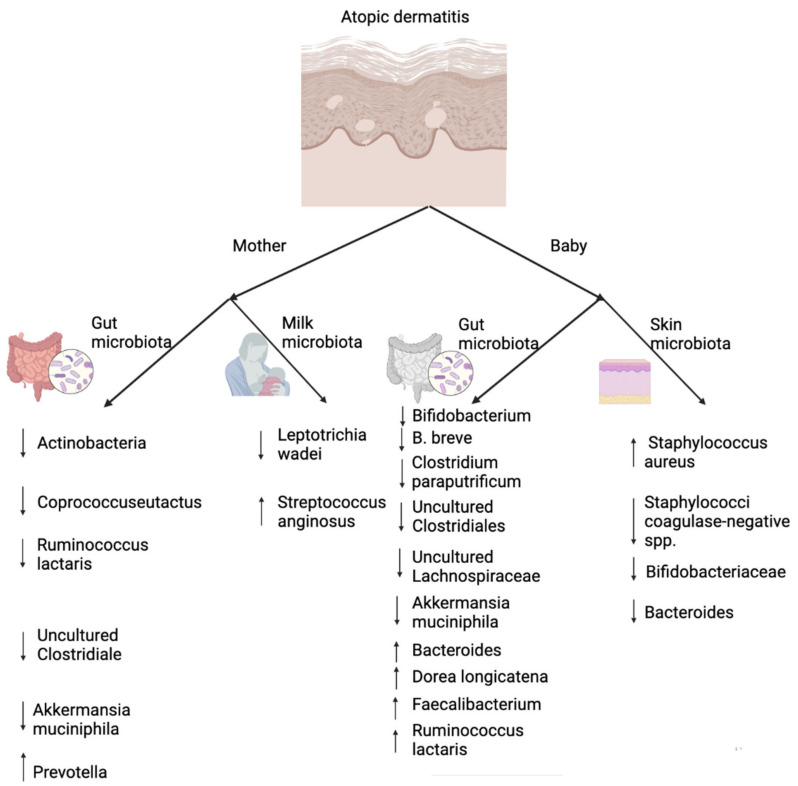
The role of maternal gut and milk microbiota and infant gut and skin microbiota in the development of AD [33,137,138,139,140,141,142]. (Arrows, upward: increased relative abundance; downward: decreased relative abundance). (Created with BioRender.com).

**Figure 5 microorganisms-10-01190-f005:**
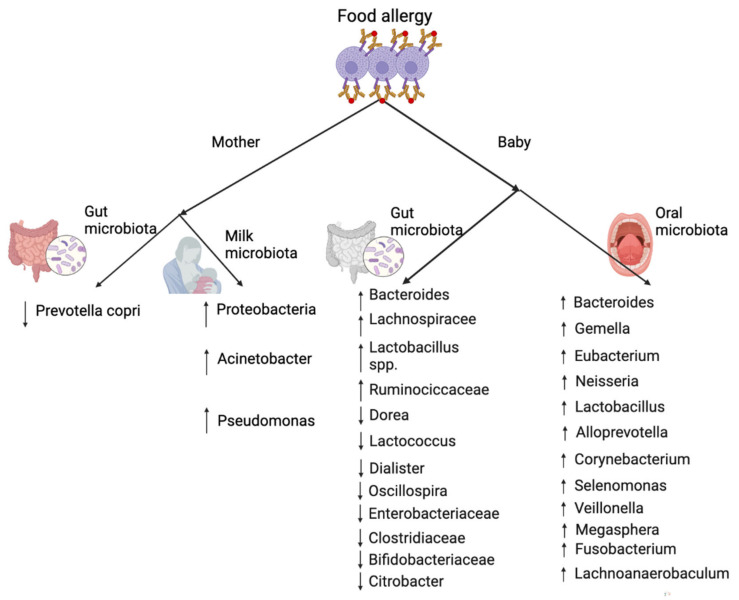
The role of maternal gut and milk microbiota and infant gut and oral microbiota in the development of FA [140,142,169,170,171]. (Arrows, upward: increased relative abundance; downward: decreased relative abundance). (Created with BioRender.com).

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
