# Peer review of "The Role of Early Life Microbiota Composition in the Development of Allergic Diseases"

_microorganisms, 2022, doi:10.3390/microorganisms10061190_

Round 1

Reviewer 1 Report

In this manuscript, Tuniyazi et al. reviewed findings on the associations between the human microbiota in the gastrointestinal tract, skin surface and respiratory tract and the development of several allergic diseases, namely asthma, atopic dermatitis and food allergy. While there are already numerous similar reviews on the topic out there, a timely update would still add value to the literature.

Major concerns:

- The title of the manuscript highlights that it is the role of the gut microbiota in the development of allergic diseases that is discussed in this manuscript; however, the authors have actually spent quite some paragraphs discussing the lung, skin (and oral) microbiota as well. Therefore, the authors should either remove the word "Gut" from the title or revise the contents accordingly.

- The authors should be more careful in differentiating causation (role) and association. An associated increase of a certain bacterial species in people with allergy does not imply that that particular bacterial species plays a role in allergy; the increase may be the result of the disease. Therefore, unless direct evidence showing a role of the microbiota is present, e.g. in mouse models, previous findings should not be used to support the claim of a role of the microbiota in allergic diseases.

- While the figures are nice in general, most of them were inserted at improper places of the text. For instance, Figure 1, which illustrates the maternal factors affecting the development of allergic diseases in infants, should not be placed in section 2, which discusses factors affecting the infant gut microbiota. Similarly, Figure 2, which displays non-maternal factors affecting the development of allergic diseases in infants, should not be placed in section 3, which discusses the role of gut microbiota in the development of immune protection.

- Potential therapeutic/preventive strategies based on microbiota modulation were discussed in the last paragraphs under each individual section of the three allergic diseases. Actually, the main concept is shared among the three diseases. Therefore, the authors are advised to form a new section focusing on this particular issue. Besides, consider adding a figure depicting this concept.

- Proper citations are missing in numerous places throughout the text. In particular, references should be provided for each of the altered bacterial taxa in Figures 3, 4 and 5, as in Figures 1 and 2. Please make sure that each factor illustrated in Figures 1 and 2 had a corresponding reference in the figure legend. Besides, when major claims are made, proper citations should always follow. 

Minor concerns:

- Genus and species names should be italicised

- P1, 3rd paragraph, 3rd line: "protozoa"

- P2, section 2: consider adding a new figure summarising the factors affecting early life gut microbiota discussed in this section.

- P2, Figure 1: consider rearranging the figure contents so that all influencing factors point to the three allergic diseases placed in the middle with arrows in different colours, representing an increase/decrease. Also for Figure 2.

- P3, 1st line: only naturally born infants have gut microbiomes similar to their mothers' vaginal microbiomes; infants born by cesarean section have gut microbiomes similar to their mothers' skin microbiome instead.

- P3, 4th line: "Streptococci"

- P3, 1st paragraph, 3rd last line: I don't think that "healthy microbiome" is specifically coined for that case.

- P5, 3rd paragraph, 3rd line line: however, Staphylococcus instead of Streptococcus was reported in Figure 3.

- English Grammar and usage need to be improved

Author Response

Response to Reviewer 1 Comments

Point 1: The title of the manuscript highlights that it is the role of the gut microbiota in the development of allergic diseases that is discussed in this manuscript; however, the authors have actually spent quite some paragraphs discussing the lung, skin (and oral) microbiota as well. Therefore, the authors should either remove the word "Gut" from the title or revise the contents accordingly.

Response 1: Thank you for your comment.

We removed the word “Gut” from the titel.

Point 2: The authors should be more careful in differentiating causation (role) and association. An associated increase of a certain bacterial species in people with allergy does not imply that that particular bacterial species plays a role in allergy; the increase may be the result of the disease. Therefore, unless direct evidence showing a role of the microbiota is present, e.g. in mouse models, previous findings should not be used to support the claim of a role of the microbiota in allergic diseases.

Response 2: Thank you for your comment.

You are right that an increase in certain microbiota may be due to many factors, including antibiotics, diet or diseases. However, in this context, microbiota is directly or indirectly related to allergic diseases. We added relative references to support the claims.

Point 3: While the figures are nice in general, most of them were inserted at improper places of the text. For instance, Figure 1, which illustrates the maternal factors affecting the development of allergic diseases in infants, should not be placed in section 2, which discusses factors affecting the infant gut microbiota. Similarly, Figure 2, which displays non-maternal factors affecting the development of allergic diseases in infants, should not be placed in section 3, which discusses the role of gut microbiota in the development of immune protection.

Response 3: Thank you for your comment.

We noticed this issue after submitting. We changed the title of section 2 into “Maternal influencing factors for the development of allergic diseases in infants

”; changed section 3 title into “Non-maternal influencing factors for the development of allergic diseases in infants”. Such changes may more accordance with the illustrations and the reviewer’s suggestions.

Point 4: Potential therapeutic/preventive strategies based on microbiota modulation were discussed in the last paragraphs under each individual section of the three allergic diseases. Actually, the main concept is shared among the three diseases. Therefore, the authors are advised to form a new section focusing on this particular issue. Besides, consider adding a figure depicting this concept.

Response 4: Thank you for your comment.

You are absolutely right. The concept based on microbiota modulation is similar in potential therapeutic/preventive strategies, including fecal, vaginal or skin microbiota transplantation, probiotic or short chain fatty acid supplement, etc. 

However, we think placing such discussions in the last paragraphs under each individual section could be better. Because, the same microbial modulation method could be aimed for different outcomes in various situations. For example, treating AD or asthma by FMT is expected different results. So, we hope you could allow us maintain the present structure.

We did try to add a picture as overview in conclusion part. However, it became too complex, as mentioned above, the same method is expected to have different results in different diseases.

Point 5: Proper citations are missing in numerous places throughout the text. In particular, references should be provided for each of the altered bacterial taxa in Figures 3, 4 and 5, as in Figures 1 and 2. Please make sure that each factor illustrated in Figures 1 and 2 had a corresponding reference in the figure legend. Besides, when major claims are made, proper citations should always follow. 

Response 5: Thank you for your comment.

We added relevant citations in Figure 3, Figure 4, and Figure 5.

Point 6: - Genus and species names should be italicised

Response 6: Thank you for your comment.

We italicised all Genus and species names.

Point 7: P1, 3rd paragraph, 3rd line: "protozoa"

Response 7: Thank you for your comment.

We corrected the word to protozoa.

Point 8: P2, section 2: consider adding a new figure summarising the factors affecting early life gut microbiota discussed in this section.

Response 8: Thank you for your comment.

We considered adding a new figure, but as we changed the title of section 2, we are hoping to write a new paper about the factors affecting early life gut microbiota in the future in a more comprehensive matter.

Point 9: P2, Figure 1: consider rearranging the figure contents so that all influencing factors point to the three allergic diseases placed in the middle with arrows in different colours, representing an increase/decrease. Also for Figure 2.

Response 9: Thank you for your comment.

We made a picture according to your suggestion. But it came out weird. The words above and below the arrows became hard to read at first look. Of course, we will replace the pictures if you instead.   

Point 10: P3, 1st line: only naturally born infants have gut microbiomes similar to their mothers' vaginal microbiomes; infants born by cesarean section have gut microbiomes similar to their mothers' skin microbiome instead.

Response 10: Thank you for your comment.

We changed the content according to your suggestion.

Point 11: P3, 4th line: "Streptococci"

Response 11: Thank you for your comment.

We changed the word into “Streptococci” according to your suggestion.

Point 12: P3, 1st paragraph, 3rd last line: I don't think that "healthy microbiome" is specifically coined for that case.

Response 12: Thank you for your comment.

We understand that using the term ‘healthy microbiome’ is inappropriate, therefore, we changed it to ‘relatively healthier’. Because the ‘healthier microbiome’ of natural born infants is compared to C-section born babies.   

Point 13: P5, 3rd paragraph, 3rd line line: however, Staphylococcus instead of Streptococcus was reported in Figure 3.

Response 13: Thank you for your comment.

We corrected as you suggested.

Point 14: English Grammar and usage need to be improved.

Response 14: Thank you for your comment. Our manuscript underwent an English language and grammar correction by a native English speaker.

Reviewer 2 Report

Dear Authors!

Please clarify the following points:

·      -   Introduction and further in the text – microbial community or microbiota, but mot microbiome, because you don't describe genomic research

2. Factors influencing early life gut microbiota

·       -  fluid or fecal microbiota transplantation (20–22) - First mention (FMT)

·         -Enterococcus, Enterobacter, and Klebsiella (and further in the text) – genera - italics needed

4. The role of lung and gut microbiota in asthma

·        - abundance of Candia and Rhodotorula fungi - Did you mean Candida?

·       -  vital for the Treg-mediated attenuation. Did you mean “CD4+CD25+ regulatory T cell”?

·       -  Figure 3. Rothina (genus) - What did you mean?

6. The role of (oral and) gut microbiota in food allerg

·         - C-section and vaginally delivered infants - earlier only «cesarean-section» without abbreviation

·         - CMA - Did you mean “cow's milk allergy”?

Author Response

Response to Reviewer 2 Comments

Point 1: Introduction and further in the text – microbial community or microbiota, but mot microbiome, because you don't describe genomic research

Response 1: Thank you for your comment.

We changed ‘microbiome’ into microbiota or microbial communities in the entire text.

Point 2: -  fluid or fecal microbiota transplantation (20–22) - First mention (FMT)

Response 2: Thank you for your comment.

We changed ‘fecal microbiota transplantation’ to ‘fecal microbiota transplantation(FMT)’.

Point 3:  -Enterococcus, Enterobacter, and Klebsiella (and further in the text) – genera - italics needed

Response 3: Thank you for your comment.

We italicised all Genus and species names.

Point 4: - abundance of Candia and Rhodotorula fungi - Did you mean Candida?

Response 4: Thank you for your comment.

We believe ‘candia’ is right. Candia is a genus of fungi.

Point 5:   -  vital for the Treg-mediated attenuation. Did you mean “CD4+CD25+ regulatory T cell”?

Response 5: Thank you for your comment.

Yes, we did. But as this is a review paper, we used the expression ‘Treg-mediated attenuation’ instead of ‘CD4+CD25+ regulatory T cell’. However, we are happy to change if you think it would be better.

Point 6: - Figure 3. Rothina (genus) - What did you mean?

Response 6: Thank you for your comment.

Rothina is a genus of fungi. Therefore, we added genus.

Point 7:  - C-section and vaginally delivered infants - earlier only «cesarean-section» without abbreviation

Response 7: Thank you for your comment.

We are sorry for this mistake. We changed ‘C-section’ to ‘cesarean-section’.

Point 8: - CMA - Did you mean “cow's milk allergy”?

Response 8: Thank you for your comment.

Yes, we did. From the perspective of gut microbiota, a previous study indicated that infants with cow's milk allergy (CMA) had…..

Round 2

Reviewer 1 Report

I do not have any further comments.